# A New 1,3-Benzodioxole Compound from *Hypecoum erectum* and Its Antioxidant Activity

**DOI:** 10.3390/molecules27196657

**Published:** 2022-10-07

**Authors:** Ning Xu, Wenli Bao, Jiletu Xin, Hua Xiao, Jiaqi Yu, Liang Xu

**Affiliations:** Inner Mongolia Key Laboratory of the Natural Products Chemistry and Functional Molecular Synthesis, Inner Mongolia Minzu University, Tongliao 028000, China

**Keywords:** *Hypecoum erectum*, spectral analysis, 1,3-benzodioxole, antioxidative activity

## Abstract

The purpose of this study was to identify the chemical components in aerial parts of *Hypecoum erectum*. A new 1,3-benzodioxole derivative, identified as Hypecoumic acid (**1**), was isolated, together with the three known compounds: protopine (**2**), coptisine (**3**), and cryptopine (**4**). Their structures were identified based on extensive spectroscopic experiments, including nuclear magnetic resonance (NMR) and high-resolution electrospray ionization mass spectra (HR-ESI-MS), as well as comparison with those reported in the literature. Meanwhile, the in vitro antioxidative activity of all compounds was determined using a DPPH-scavenging assay, and compound 1 (IC_50_ = 86.3 ± 0.2 μM) was shown to have moderate antioxidative activity.

## 1. Introduction

The oxidative–antioxidant balance affects the proper functioning of homeostasis. Overproduction of reactive oxygen species (ROS) causes oxidative stress, and is one of the most common reasons for homeostasis disorders. The antioxidative activities of herbal medicines have received a great amount of attention as being primary preventive ingredients against various diseases, such as cancer [1], diabetes [2], and cardiovascular diseases [3]. *H. erectum*, belonging to the Papaveraceae family, is widely distributed in North, Northeast, and Northwest China [4]. *H. erectum* has traditionally been used to treat inflammation, fever, and pain in folk medicine in China. The chemical constituents of *H. erectum* L. are mainly isoquinoline alkaloids, such as hypecorine and hypecorinine [5], hyperectine [6], protopine, coptyine and allocryptopine [7], isohyperectine [8], 2,3-Dimethoxy-*N*-formylcorydamine and 2,3-Dimethoxyhypecorinine [9], and leptocarpinine [10]. Recent pharmacological studies involving in vitro experiments have been performed on the anti-inflammatory [9], antimicrobial [11], antibacterial and analgesic activity [12] of *H. erectum*. However, few comprehensive studies have investigated its in vivo antioxidative abilities and identified the bioactive metabolites. In preliminary experiments, we found that the extract of aerial parts of *H. erectum* showed antioxidative activity, and therefore set out to identify the bioactive compounds. Herein, we report on the isolation and structural elucidation of a new 1,3-benzodioxole derivative (**1**), along with three known compounds (**2**–**4**) (Figure 1).

## 2. Results

### 2.1. Structure Elucidation of Isolated Compound

Compound **1**, which was obtained as a yellow powder, was assigned the molecular formula C_13_H_14_O_6_ on the basis of an HRESIMS (positive-ion mode) ion at *m*/*z* 267.0860 [M + H]^+^ (calculated for C_13_H_15_O_6_, 267.0868) and NMR data (Table 1 and Appendix A). Its infrared and ^13^C NMR spectra displayed the carboxyl functional group (1793 and 3566 cm^−1^; 169.5 ppm). Analysis of the ^1^H NMR spectra of compound **1** indicated the presence of a 1,2,3,4-tetrasubstituted benzene moiety, as determined by the *δ*_H_ 6.89 (d, *J* = 8.3 Hz) and *δ*_H_ 7.66 (d, *J* = 8.3 Hz) signals. The ^1^H-^1^H COSY correlations, starting at H-9 via H-10/H-11 and ending at H-12, were suggestive of an *n*-butoxy group (Figure 1). The ^13^C NMR data of compound **1** exhibited a total of 13 carbon resonances corresponding to six aromatic carbons, two carbonyl carbons, four methylene carbons, and one methyl carbon. Dioxymethylene was placed at C-3 and C-4 on the basis of the HMBC correlations from *δ*_H_ 6.15 (s, *δ*_C_ 102.8) to *δ*_C_ 152.1 (C-3) and *δ*_C_ 121.2 (C-4), which defined 1,3-benzodioxole moiety [12]. The HMBC correlations from *δ*_H_ 4.37 (H-9) to *δ*_C_ 165.3 (C-8), and from *δ*_H_ 7.66 (H-6) to *δ*_C_ 121.2 (C-4) indicated that the *n*-butoxy group was attached to *δ*_C_ 165.3 (C-8), connecting to the benzene ring. HMBC cross-peaks observed between *δ*_H_ 7.66 (H-6) and *δ*_C_ 169.5 (C-13), as well as between *δ*_H_ 6.89 (H-1) and *δ*_C_ 117.0 (C-5), placed the carboxyl group at C-5. Thus, the structure of compound **1** was established to be 4-(butoxycarbonyl)benzo[d][1,3]dioxole-5-carboxylic acid, and it was named Hypecoumic acid (**1**) (Figure 2).

*H. erectum* samples were extracted successively with 95% EtOH, which was then subjected to chromatography to yield four compounds. On the basis of the reported MS and NMR data, the following known compounds were identified: protopine (**2**) [13], coptisine (**3**) [14], and cryptopine (**4**) [15].

### 2.2. DPPH-Scavenging Assay

The DPPH is a stable free radical, which accepts the hydrogen radical or an electron, forming a stable diamagnetic molecule [13]. This antioxidant ability can be assessed by the determination of IC_50_ values related to the amount of the sample required to reduce 50% of free radicals. The four compounds were further investigated using the DPPH-scavenging assay to evaluate antioxidant activity (Table 2). The most effective antioxidant was found to be compound **1,** with an IC50 of 86.3 μM, followed by coptisine (**3**), protopine (**2**), and cryptopine (**4**) with values of IC_50_ 252.6, 345.2, and 430.1 ± 1.6 μM, respectively.

## 3. Materials and Methods

### 3.1. General Experimental Procedures

Methanol, ethanol, petroleum ether, ethyl acetate, n-butanol, and distilled water were used for extraction, fractionation, and open column chromatography. Chloroform-d (CDCl_3_-d) and tetramethylsilane, used as the internal standard, were purchased from Sigma-Aldrich Co. (Sigma-Aldrich Co., St. Louis, MO, USA) and as solvents for NMR analysis. Silica gel (200–300 mesh; Qingdao Haiyang Chemical Co., Ltd., Qingdao, China) and Sephadex LH-20 resin (25–100 μm; Pharmacia, Uppsala, Sweden) were used for open-column chromatography. Precoated silica gel GF-254 glass plates (Qingdao Haiyang Chemical Co., Ltd., Qingdao, China) were used for the TLC. The compounds were visualized under UV (254 nm) light and by spraying with H_2_SO_4_-EtOH (1:9, *v*/*v*), followed by heating.

The HRESIMS data were acquired on a Q Exactive Focus mass spectrometer (Thermo Fisher Scientific, Dreieich, Germany). UV data were recorded using a UV-670 Double Beam spectrometer (Mapada, Shanghai, China). IR spectra were obtained with an IRAffinity-1S spectrophotometer (Shimadzu, Kyoto, Japan). NMR spectra were recorded on a Bruker Avance spectrometer (Billerica, MA, USA) operating at 500 (^1^H) and 125 (^13^C) MHz, with tetramethylsilane used as the internal standard. Semi-preparative HPLC was performed on a Shimadzu LC-6AD HPLC system with a reversed-phase C_18_ column (10 × 250 mm, 10 μm) (Shimadzu, Kyoto, Japan).

### 3.2. Plant Material

*H. erectum* specimens were purchased from the Affiliated Hospital, Inner Mongolia Minzu University, Tongliao, China, in May 2017 and identified by Prof. Xi Quan of the Inner Mongolia University for Nationalities. A sample (No. 20170613) was deposited at the Inner Mongolia Minzu University.

### 3.3. Extraction and Isolation

Aerial parts of *H. erectum* (12.0 kg) were extracted with 95% EtOH under reflux (3 × 60 L, 2.0 h). The solvents were removed in vacuo using a rotary evaporator. The crude extract was suspended in distilled water and partitioned with petroleum ether, EtOAc, and *n*-BuOH, yielding 51.0, 78.0 and 733.0 g of residue, respectively. The *n*-BuOH extract (733.0 g) was then subjected to D101 macroporous adsorption resin, using 30% EtOH, 60% EtOH and EtOH as the eluent, yielding three fractions (Fr. 1–3). Fr 2 (54.0 g) was subjected to silica gel column chromatography (CH_2_Cl_2_–MeOH, 80:1–0:1, *v*/*v*), yielding ten subfractions (Fr. 2.1–2.10). Fr. 2.1 (300.6 mg) was subsequently purified on semi-preparative HPLC (CH_3_CN–H_2_O, 20:80–95:5; flow rate: 2.5 mL/min; 254 nm) to obtain compound **1** (5.6 mg, t_R_ = 9.4 min), which was further purified by a Sephadex LH-20 column (CHCl_3_-MeOH, 1:1, *v*/*v*). Fr. 2.3 (500.7 mg) was subjected to semi-preparative HPLC (CH_3_CN–H_2_O, 20:80–95:5; flow rate: 2.5 mL/min; 320 nm) to yield eight subfractions (Fr. 2.2.1–2.2.8). Fr. 2.2.5 (68.5 mg) was purified by silica gel column chromatography (CH_2_Cl_2_–MeOH, 150:1–1:1, *v*/*v*), yielding compounds **2** (8.1 mg) and **3** (9.5 mg). Additionally, Fr. 2.5 (103.4 mg) was separated into six subfractions (Frs. 2.5.1–2.5.6) by Sephadex LH-20 chromatography (CHCl_3_-MeOH, 1:1, *v*/*v*), following which Fr. 2.5.3 was further purified by preparative HPLC (MeOH–H_2_O, 35:65; flow rate: 2.5 mL/min; 254 nm) to yield compound **4** (4.5 mg, t_R_ = 8.8 min).

### 3.4. Compound Characterization Data

Hypecoumic acid: yellow powder (MeOH). Infrared (KBr) νmax: 3366, 2954, 2897, 1793 and 1584 cm^−1^. HRESIMS at m/z 267.080 [M + H]^+^ (calculated for 267.086). For ^1^H NMR (500 MHz, CDCl_3_) and ^13^C NMR (125 MHz, CDCl_3_) data, see Table 1.

### 3.5. DPPH-Scavenging Assay

The DPPH (2,2-diphenyl-1-picrylhydrazyl) reagent (at a concentration of 0.1 μM, dissolved in MeOH) was used for assaying the reduction of free radicals by the various compounds, as measured by UV spectrometry [16,17]. Compounds **1**–**4** were dissolved in MeOH to obtain a suitable concentration (100–150 μg/mL), and were dispensed into a cuvette containing 3.0 mL of DPPH solution. The resultant mixture was incubated for 30 min at ambient temperature in the dark and then monitored at 517 nm. The DPPH-scavenging activity was calculated according to the following equation:DPPH-scavenging activity (%) = (A_C_ − A_S_)/A_C_ × 100(1)
where A_C_ is the initial concentration of the stable DPPH free radical without the test compound, and A_S_ is the absorbance of the remaining concentration of DPPH in the presence of MeOH. The IC_50_ values (μM) were determined from a plotted graph of DPPH-scavenging activity against the concentrations of the respective compounds, where IC_50_ is defined as the total amount of antioxidant needed to decrease the initial DPPH free-radical concentration by 50% (mean ± SD, *n* = 3).

## 4. Conclusions

In this study, the crude extract of *H. erectum* revealed the isolation and purification of one new and three known compounds. Compound **1** was clarified to be 4-(butoxycarbonyl)benzo[d][1,3]dioxole-5-carboxylic acid, named Hypecoumic acid. This is the first report of the isolation of a 1,3-Benzodioxole compound from the Papaveraceae family. All the isolates were evaluated for their antioxidative activity. Among them, Hypecoumic acid (**1**) exhibited moderate capabilities against ·OH. These findings enrich the knowledge of the chemical diversity and biological potential of *H. erectum.*

## Figures and Tables

**Figure 1 molecules-27-06657-f001:**
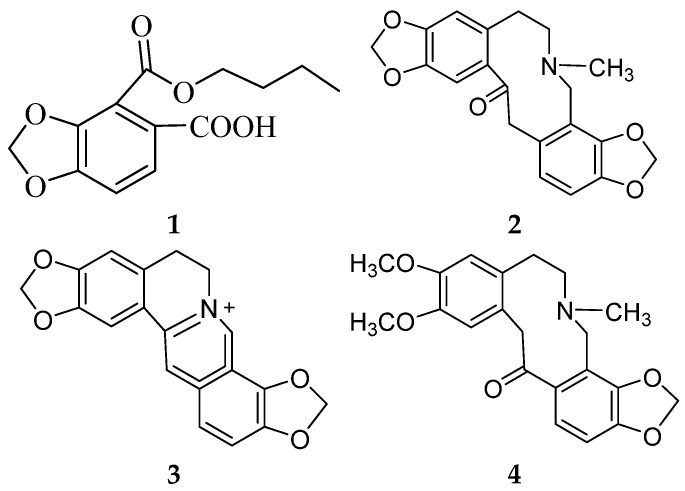
Structures of the compounds (**1**–**4**).

**Figure 2 molecules-27-06657-f002:**
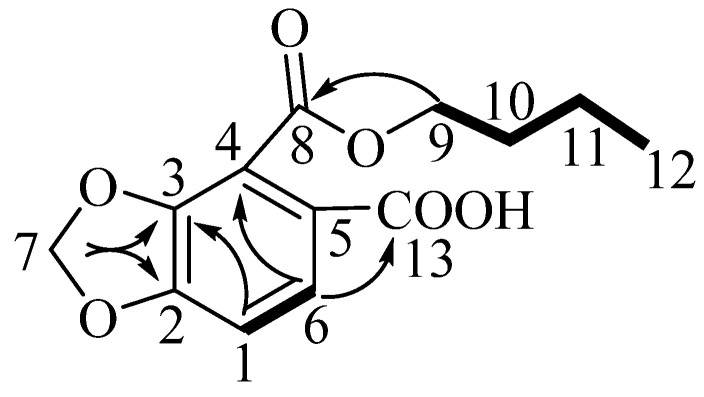
Key HMBC and ^1^H-^1^H COSY correlations for compound **1**.

**Table 1 molecules-27-06657-t001:** ^1^H and ^13^C-NMR data of compound **1** in CDCl_3_-d (*δ* in ppm, *J* in Hz).

	*δ* _H_	*δ* _C_
1	6.89 (1H, d, 8.3)	108.7
2	-	146.0
3	-	152.1
4	-	121.2
5	-	117.0
6	7.66 (1H, d, 8.3)	126.6
7	6.15 (1H, s)	102.8
8	-	165.3
9	4.37 (2H, t, 6.5)	66.0
10	1.75 (2H, m)	30.4
11	1.46 (2H, m)	19.1
12	0.98 (3H, t, 7.42)	13.7
13	-	169.5

**Table 2 molecules-27-06657-t002:** Antioxidant activity of compounds **1**–**4**.

Name	IC_50_ Value (μM)
Hypecoumic acid	86.3 ± 0.2
protopine	345.2 ± 1.3
coptisine	252.6 ± 2.1
cryptopine	430.1 ± 1.6

## Data Availability

The data used to support the findings of this study are available from the corresponding author upon request.

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
