# Peer review of "A New 1,3-Benzodioxole Compound from Hypecoum erectum and Its Antioxidant Activity"

_molecules, 2022, doi:10.3390/molecules27196657_

Round 1

Reviewer 1 Report

Manuscript presented by Ning Xu  et al. shows a study about new 1,3-Benzodioxole Compound from Hypecoum Erectum and its antioxidant activity. The manuscript is written and prepared not very neatly. Several aspects should be improved (or need to be re-written).

I recommend the article to publish but first the paper should be improve. My decision – Reconsider after major revision. Comments to be considered, in order to further improve the manuscript quality:

(1) Introduction need to be re-written, the information flow of introduction is poor.

(2) Avoid lumping references as in 2-4 as well as 5-9 and all other. Instead summarise the main contribution of each referenced paper in a separate sentence.

(3) The abstract and conclusion are too short and general. Please correct and improve the section.

(4) More informations in “Materials and Methods” section are necessary, eg. NMR solvent, internal standard ect.

(5) Add “1,3-benzodioxole” and “spectral analysis” to keywords.

(6) In order to show that the topic of presented manuscript is proper for the publication in MOLECULES please include into reference other publications of this journal.

(7) The style of manuscript (especially reference) should be improve (see template).

(8) Superscripts and subscripts as well as commas and periods should be cahnge and correct. Avoid extra spaces and enters. It is necessary to publish article.

(9) The English correction is necessary.

Presented comments are not everything aspects. Please re-fresh whole manuscript.

Author Response

Dear Reviewer

  Thanks very much for your kind work and consideration on publication of our paper. On behalf of my co-authors, we would like to express our great appreciation to editor and reviewers. We have studied the your comments carefully and have made correction, which we hope meet with approval. Revised portion are marked in red in the paper. The main corrections in the paper and the responds to the reviewer' comments are as flowing:

1. Introduction need to be re-written, the information flow of introduction is poor.Avoid lumping references as in 2-4 as well as 5-9 and all other. Instead summarise the main contribution of each referenced paper in a separate sentence.

Response to comment: Thank you for your careful review. In the revised manuscript, we have made correction according to the reviewer’s comments, and they are marked in red in the paper, in the line 22-39, page 1.

2. The abstract and conclusion are too short and general. Please correct and improve the section.

Response to comment: Thank you for your careful review. We have made correction according to the reviewer’s comments as the following: “The purpose of this study was to identify the chemical components in aerial parts of Hypecoum erectum. A new 1,3-benzodioxole derivative, identified as Hypecoumic acid (1), was isolated together with the three known compounds: protopine (2), coptisine (3), and cryptopine (4). Their structures were identified based on extensive spectroscopic experiments, including nuclear magnetic resonance (NMR), and high-resolution electrospray ionization mass spectra (HR-ESI-MS), as well as the comparison with those reported in the literature. Meanwhile, the in vitro Antioxidative activity of all compounds was determined using DPPH-scavenging assay, and compounds 1 (IC50 = 86.3 ± 0.2 µM) were shown to have moderate antioxidative activity.” in the line 10-17, page 1, and “In this study, The crude extract of Hypecoum erectum revealed the isolation and purification of one new and three known compounds. Compound 1 was clarified to be 4-(butoxycarbonyl)benzo[d][1,3]dioxole-5-carboxylic acid, named Hypecoumic acid. This is the first report of the isolation of 1,3-Benzodioxole Compound from the Papaveraceae family. All the isolates were evaluated for their antioxidative activity. Among them, Hypecoumic acid (1) exhibited moderate capabilities against ·OH. These findings enrich the knowledge of the chemical diversity and biological potential of Hypecoum erectum.” in the line 157-164, page 4.

3. More informations in “Materials and Methods” section are necessary, eg. NMR solvent, internal standard ect.

Response to comment: Thank you for your careful review. We have made correction according to the reviewer’s comments as the following: “Chloroform-d (CDCl3-d) with tetramethylsilane used as the internal standard were purchased from Sigma-Aldrich Co. (Sigma-Aldrich Co., St. Louis, MO, USA) as solvents for NMR analysis. ” in the line 95-104, page 3.

4. Add 1,3-benzodioxole” and “spectral analysis” to keywords.

Response to comment: Thank you for your careful review. We have made correction according to the reviewer’s comments as the following: “Keywords: Hypecoum erectum; spectral analysis; 1,3-benzodioxole; Antioxidative activity ” in the line 18, page 1.

5. In order to show that the topic of presented manuscript is proper for the publication in Molecules please include into reference other publications of this journal.

Response to comment: Thank you for your careful review. We have made correction according to the reviewer’s comments as the following: “17 Tong, Z.; Xiao X.Y.; Lu, Y.Y.; Zhang, Y.X.; Hu, P.; Jiang, W.; Zhou, H.; Pan S.X.; Huang, Z.Y.; Hu, L.Z. New Metabolites from Aspergillus Ochraceus with Antioxidative Activity and Neuroprotective Potential on H2O2 Insult SH-SY5Y Cells. Molecules. 2022, 27, 52. ” in the line 213-215, page 5.

6. The style of manuscript (especially reference) should be improve (see template).

Response to comment: Thank you for your careful review. In the revised manuscript, we have made correction according to the reviewer’s comments, and they are marked in red in the paper, in the line 182-215, page 4.

7. Superscripts and subscripts as well as commas and periods should be change and correct. Avoid extra spaces and enters. It is necessary to publish article.

Response to comment: Thank you for your careful review. In the revised manuscript, we have made correction according to the reviewer’s comments, and they are marked in red in the paper.

8. The English correction is necessary.

Response to comment: Thank you for your careful review. the revised manuscript have corrected by native English man.

Reviewer 2 Report

The introduction is poor and must be improved showing more importance of this plant and confirming if other authors identified those compounds because in this text it is not clear.

The botanical name of the specie must be in the italic form in all text.

For ethanolic extraction, authors must include the solid:solvent ratio used to obtain the extract considering all successive extractions. Please, also indicate the extraction replicates that the authors did, to guarantee that these compounds are presented in a large sample.

The discussion should be improved since it seems a chemical report.

For values of IC50, there is no error or standard deviation in all results. How many replicates were made?

For figure 2, please present the correlations for all compounds.

There is missing Table 1.

There are no results presented for HPLC and TLC analyses. Please, provide results from HPLC and TLC used methods, and improve the quality of images from supplementary results.

For chromatographic results, please present the retention time of those compounds and concentrations obtained and complete the information of gradient elution for fractionations including the time of each programming, the flow rate of solvents and the wavelength of maximum absorption for these isolated compounds.

For TLC describe the solvent used in the elution of the plate and the retention time for selected compounds. Please, provide an image of TLC.

Section 3.4 there is missing table 1 or the number of Table 2 is incorrect.

Section 3.5 authors must indicate the number of replicates used in this assay and present the variation of this result. Only the result of compound 1 presents the variation, and it could not be equal for all.

Author Response

Dear Reviewer

  Thanks very much for your kind work and consideration on publication of our paper. On behalf of my co-authors, we would like to express our great appreciation to editor and reviewers. We have studied the your comments carefully and have made correction, which we hope meet with approval. Revised portion are marked in red in the paper. The main corrections in the paper and the responds to the reviewer' comments are as flowing:

1. The introduction is poor and must be improved showing more importance of this plant and confirming if other authors identified those compounds because in this text it is not clear.

Response to comment: Thank you for your careful review. In the revised manuscript, we have made correction according to the reviewer’s comments, and they are marked in red in the paper, in the line 22-39, page 1.

2. The botanical name of the specie must be in the italic form in all text.

Response to comment: Thank you for your careful review. In the revised manuscript, we have made correction according to the reviewer’s comments, and they are marked in red in the paper.

3. For ethanolic extraction, authors must include the solid:solvent ratio used to obtain the extract considering all successive extractions. Please, also indicate the extraction replicates that the authors did, to guarantee that these compounds are presented in a large sample.

Response to comment: Thank you for your careful review. We have made correction according to the reviewer’s comments as the following: “Aerial parts of H. erectum (12.0 kg) were extracted with 95% EtOH under reflux (3 x 60 L, 2.0 h). ” in the line 120-121, page 4.

4. The discussion should be improved since it seems a chemical report.

Response to comment: Thank you for your careful review. In the revised manuscript, we have made correction according to the reviewer’s comments in the page 2-3.

5. For values of IC50, there is no error or standard deviation in all results. How many replicates were made?

Response to comment: Thank you for your careful review. In the revised manuscript, we have made correction according to the reviewer’s comments in the line 75-84, the page 3.

6. For figure 2, please present the correlations for all compounds.

Response to comment: Thank you for your careful review. We feel very sorry, and could not provide the correlations for all compounds. But Compound 2-4 were identified based on 1D NMRand HR-ESI-MS, as well as the comparison with those reported in the literature.

7. There is missing Table 1.

Response to comment: Thank you for your careful review. We have made correction according to the reviewer’s comments as the following: “Table 1. 1H and 13C-NMR data of compound 1 in CDCl3-d (δ in ppm, J in Hz).” in the line 72, page 2.

8. There are no results presented for HPLC and TLC analyses. Please, provide results from HPLC and TLC used methods, and improve the quality of images from supplementary results.

Response to comment: Thank you for your careful review. We feel very sorry, and could not provide results from TLC used methods, because image were not saved. But TLC doesn't affect the results of the experiment. In supplementary Information, We have made correction according to the reviewer’s comments.

Figure S1. 1H-NMR (500 MHz, CDCl3-d) spectrum of compound 1

Semi-preparative HPLC chromatograms of Fr. 2.1

 Semi-preparative HPLC chromatograms of Fr. 2.5

9. For chromatographic results, please present the retention time of those compounds and concentrations obtained and complete the information of gradient elution for fractionations including the time of each programming, the flow rate of solvents and the wavelength of maximum absorption for these isolated compounds.

Response to comment: Thank you for your careful review. In the revised manuscript, we have made correction according to the reviewer’s comments, and they are marked in red in the paper, in the line 120,123 and 128, page 4.

10. For TLC describe the solvent used in the elution of the plate and the retention time for selected compounds. Please, provide an image of TLC.

Response to comment: Thank you for your careful review. We feel very sorry, and could not provide results from TLC used methods, because image were not saved. But TLC doesn't affect the results of the experiment.

11. Section 3.4 there is missing table 1 or the number of Table 2 is incorrect.

Response to comment: Thank you for your careful review. We have made correction according to the reviewer’s comments as the following: “Table 1. 1H and 13C-NMR data of compound 1 in CDCl3-d (δ in ppm, J in Hz).” in the line 72, page 2.

12. Section 3.5 authors must indicate the number of replicates used in this assay and present the variation of this result. Only the result of compound 1 presents the variation, and it could not be equal for all.

Response to comment:Thank you for your careful review. In the revised manuscript, we have made correction according to the reviewer’s comments as the following: “The IC50 values (μM) were determined from a plotted graph of DPPH-scavenging activity against the concentrations of the respective compounds, where IC50 is defined as the total amount of antioxidant needed to decrease the initial DPPH free radical concentration by 50% (mean ± SD, n = 3).” in the line 147, page 4.

Reviewer 3 Report

This manuscript reports the isolation of one new and three known compounds from Hypecoum erectum as well as their antioxidant activities. The experimental methods and the structural elucidation of the new compound seem to be appropriate, and the results are valuable as a description of a new natural product.

The following minor points should be considered.

1. The name of the compound "Hypecoum A" is taken from the genus name of the plant from which it is derived. This is not a common way of naming natural products. In general, compound names are based on the organism names but not the same (e.g., hyppecoumin A or hypecoumic acid in this case).

2. The names of the organism should be italicized in the title and abstract.

3. Table. 2

J values should be given for the triplet signals (H-9 and H-12). The multiplicity of carbons should also be listed.

4. Line 90-92

According to this sentence, Fr. 2-1 was first separated by HPLC and then purified by the Sephadex LH-20 column. Is it right? Usually, HPLC is used at the final step.

Author Response

Dear Reviewer

Thanks very much for your kind work and consideration on publication of our paper. On behalf of my co-authors, we would like to express our great appreciation to editor and reviewers. We have studied the your comments carefully and have made correction, which we hope meet with approval. Revised portion are marked in red in the paper. The main corrections in the paper and the responds to the reviewer' comments are as flowing:

1. The name of the compound "Hypecoum A" is taken from the genus name of the plant from which it is derived. This is not a common way of naming natural products. In general, compound names are based on the organism names but not the same (e.g., hyppecoumin A or hypecoumic acid in this case).

Response to comment: Thank you for your careful review. We have made correction according to the reviewer’s comments as the following: “Hypecoumic acid” in the revised manuscrip.

2. The names of the organism should be italicized in the title and abstract.

Response to comment: Thank you for your careful review. In the revised manuscript, we have made correction according to the reviewer’s comments, and they are marked in red in the paper.

3. Jvalues should be given for the triplet signals (H-9 and H-12). The multiplicity of carbons should also be listed.

Response to comment: Thank you for your careful review. In the revised manuscript, we have made correction according to the reviewer’s comments in the page 2.

Table 1. 1H and 13C-NMR data of compound 1 in CDCl3-d (δ in ppm, J in Hz).

δH

δC

1

6.89 (1H, d, 8.3)

108.7

2

-

146.0

3

-

152.1

4

-

121.2

5

-

117.0

6

7.66 (1H, d, 8.3)

126.6

7

6.15 (1H, s)

102.8

8

-

165.3

9

4.37 (2H, t, 6.5)

66.0

10

1.75 (2H, m)

30.4

11

1.46 (2H, m)

19.1

12

0.98 (3H, t, 7.42)

13.7

13

-

169.5

4. Line 90-92, According to this sentence, Fr. 2-1 was first separated by HPLC and then purified by the Sephadex LH-20 column. Is it right? Usually, HPLC is used at the final step.

Response to comment: Thank you for your careful review. Compoud 1 obtained by semi-preparative HPLC was not pure, and then was further purified by a Sephadex LH-20 column.

Round 2

Reviewer 1 Report

Authors have satisfactorily answered all my comments. Despite some minor editorial errors still present, the manuscript can be accepted. Before publication of article the improvement  is required prior.